# Longitudinal Analyses after COVID-19 Recovery or Prolonged Infection Reveal Unique Immunological Signatures after Repeated Vaccinations

**DOI:** 10.3390/vaccines10111815

**Published:** 2022-10-28

**Authors:** Daisuke Hisamatsu, Akari Ikeda, Lisa Ito, Yasushi Matsushita, Makoto Hiki, Hirotake Mori, Yoko Tabe, Toshio Naito, Chihiro Akazawa

**Affiliations:** 1Intractable Disease Research Center, Juntendo University Graduate School of Medicine, Tokyo 113-8421, Japan; 2Department of Internal Medicine and Rheumatology, Faculty of Medicine, Juntendo University School of Medicine, Tokyo 113-8421, Japan; 3Department of Cardiovascular Biology and Medicine, Juntendo University Faculty of Medicine, Tokyo 113-8421, Japan; 4Department of General Medicine, Juntendo University Faculty of Medicine, Tokyo 113-8421, Japan; 5Department of Clinical Laboratory Medicine, Juntendo University Graduate School of Medicine, Tokyo 113-8421, Japan

**Keywords:** COVID-19, cytokine, immunodeficiency, vaccine biomarker

## Abstract

To develop preventive and therapeutic measures against coronavirus disease 2019, the complete characterization of immune response and sustained immune activation following viral infection and vaccination are critical. However, the mechanisms controlling intrapersonal variation in antibody titers against SARS-CoV-2 antigens remain unclear. To gain further insights, we performed a robust molecular and cellular investigation of immune responses in infected, recovered, and vaccinated individuals. We evaluated the serum levels of 29 cytokines and their correlation with neutralizing antibody titer. We investigated memory B-cell response in patients infected with the original SARS-CoV-2 strain or other variants, and in vaccinated individuals. Longitudinal correlation analyses revealed that post-vaccination neutralizing potential was more strongly associated with various serum cytokine levels in recovered patients than in naïve individuals. We found that IL-10, CCL2, CXCL10, and IL-12p40 are candidate biomarkers of serum-neutralizing antibody titer after the vaccination of recovered individuals. We found a similar distribution of virus-specific antibody gene families in triple-vaccinated individuals and a patient with COVID-19 pneumonia for 1 year. Thus, distinct immune responses occur depending on the viral strain and clinical history, suggesting that therapeutic options should be selected on a case-by-case basis. Candidate biomarkers that correlate with repeated vaccination may support the efficacy and safety evaluation systems of mRNA vaccines and lead to the development of novel vaccine strategies.

## 1. Introduction

Therapeutic options for coronavirus disease 2019 (COVID-19), caused by severe acute respiratory syndrome coronavirus 2 (SARS-CoV-2) infection, include convalescent plasma therapy, antiviral drugs, and monoclonal antibody therapy. However, these strategies have not changed since their development in 2019, when the original SARS-CoV-2 strain (defined here as wild-type) emerged, despite the fact that multiple mutations have since been recorded. A major clinical challenge in treating patients with severe COVID-19 is hyperactivation of the immune system, which is caused by a “cytokine storm”, that is, the overproduction of systemic pro-inflammatory cytokines [1,2]. Cytokine storms are thought to cause multiorgan failure, but their mechanism of onset is not well understood. As of April 2022, five variants of concern (VOCs) have been identified by the World Health Organization: Alpha (also known as B.1.1.7), Beta (B.1.351), Gamma (P.1), Delta (B.1.617.2), and Omicron (B.1.1.529). These variants exhibit increased infectivity and pathogenicity along with reduced sensitivity to neutralizing antibodies in comparison with the wild-type (WT) SARS-CoV-2 strain [3,4,5,6,7,8]. Mutations within VOCs are likely responsible for the varied immune responses induced by infection. Candidates include mutations in the Spike (S) protein and its receptor-binding domain (RBD), which may affect binding to human angiotensin-converting enzyme 2 (ACE2) and signaling pathways induced upon cellular entry [9]. Therefore, different SARS-CoV-2 strains may induce different cytokine responses, such that specific cytokine profiles may be associated with clinical characteristics.

Adequate longitudinal protection against severe COVID-19 requires more than one dose of the common SARS-CoV-2 vaccines [10]. Several studies have found differences in the degree and duration of neutralizing antibody responses induced by different vaccines, which include messenger RNA (mRNA)-based vaccines from Pfizer-BioNTech (BNT162b2) and Moderna (mRNA-1273), and in the responses induced by subsequent booster vaccinations [10,11,12,13]. Furthermore, a study in recovered COVID-19 patients showed that a rapid antigen-specific memory B-cell (Bmem) response occurs after the first vaccine dose, with serum antibody titers reaching a peak value [11,14,15,16,17]. However, biomarkers to guide the repeated vaccine are unknown, especially in recovered individuals.

The acquisition of high-affinity antibodies, a process known as affinity maturation, occurs via diversification and clonal selection through somatic hypermutations (SHMs) introduced during the active proliferation of antigen-stimulated B-cells in the germinal center [18]. In recovered COVID-19 patients, the accumulation of SHMs in antibody genes induces the formation of potent and broadly neutralizing antibodies that are effective against VOCs [19,20]. Compared with naïve individuals, infected individuals who receive repeat vaccinations have been reported to exhibit a more potent antibody response to VOCs [21]. By contrast, some immunocompromised individuals receiving immunosuppressive therapy such as rituximab, fail to develop anti-SARS-CoV-2 antibodies at the symptom onset and following repeated vaccinations [22]. However, the molecular basis of these immunological processes and the B-cell receptor (BCR) repertoire after infection or vaccination remains unclear, particularly in patients with immunodeficiency.

Approximately 100 d after infection, most individuals experience a decrease in serum antibody levels against SARS-CoV-2 [23,24]. Even after vaccination, the serum antibody titers are low in some populations, such as immunocompromised patients and elderly participants [25,26]. Furthermore, the emergence of VOCs is speculated to maintain and encourage human-to-human transmission and cause residual infections in immunocompromised hosts [27,28,29,30]. Choi et al. performed a 152-day longitudinal analysis of a patient receiving immunosuppressive therapies and showed that amino acid changes were observed in 57% of S genes and 38% of RBD genes [31]. Furthermore, late-phase mutations observed in this patient were resistant to a common class of neutralizing antibodies derived from healthy recovered COVID-19 patients and clinical monoclonal antibodies [31]. In such immunocompromised hosts with prolonged infection, SARS-CoV-2 is thought to evolve into a new strain that is more proficient at immune evasion [29,30,32,33]. We thought that antibodies produced against the new strain carrying an unknown mutation might be helpful as a therapeutic monoclonal antibody.

In this study, we aimed to determine whether the immune response induced by SARS-CoV-2 infection or vaccination is affected by infection strain and/or the presence of underlying comorbidities. Furthermore, we sought to identify molecular biomarkers in individuals vaccinated against SARS-CoV-2, as these may help in the management of longitudinal dosing. We hypothesized that a unique Bmem response occurs in immunocompromised patients, generating broadly neutralizing antibodies against new or unknown variants. To this end, we performed longitudinal analyses of cytokine profiles associated with neutralizing antibody activity, including the distribution of antibody gene sequences in patients with or without comorbidities during infection and after vaccination. Our work shows that sequencing of antibody genes after recovery of immunocompromised patients may reveal antibody immunogenicity. Furthermore, the antibodies produced by such immunocompromised patients are expected to be valuable as therapeutic monoclonal antibodies.

## 2. Methods

### 2.1. Study Design

Patients were recruited at the Juntendo University Hospital in Tokyo, Japan, between August 2020 and January 2021. Study participants were adults aged 21–86 years diagnosed with COVID-19 by using real-time RT-PCR for nasopharyngeal swabs using LightMix Modular SARS-CoV (COVID-19) N-gene and E-gene assays (Roche Diagnostics, Tokyo, Japan) according to manufacturer instructions. The severity of COVID-19 was classified according to the Japan Ministry of Health, Labor, and Welfare criteria. Briefly, mild cases were defined by mild clinical symptoms such as cough and fever without pneumonia, according to imaging examinations. The moderate cases were divided into two categories: moderate I was defined by mild respiratory symptoms with pneumonia according to imaging examination, and moderate II was defined by pneumonia on imaging with respiratory insufficiency and oxygen saturation ≤93% in a resting state. Healthy controls were volunteers who visited Juntendo University Hospital between August and December 2020 and provided a negative RT-PCR test before blood collection. Blood samples were collected at admission and discharge, and convalescent serum samples were collected during outpatient follow-up after discharge.

### 2.2. Measurement of Anti-SARS-CoV-2 S-RBD IgM and IgG antibodies

The levels of anti-SARS-CoV-2 S-RBD-specific IgM and IgG antibodies in serum samples were determined using the IgM for SARS-CoV-2 S-RBD ELISA Kit (Proteintech, Rosemont, IL, USA) and the anti-SARS-CoV-2 S-RBD protein Human IgG ELISA Kit (Proteintech) according to manufacturer instructions. All serum samples were diluted 1:100 and assays were performed in duplicate.

### 2.3. In Vitro Neutralization Assay 

The following kits were used to measure the neutralizing potency against each strain following manufacturer instructions: SARS-CoV-2 Neutralization Antibody Detection Kit (Medical and Biological Laboratories, Tokyo, Japan) for the WT SARS-CoV-2 strain, SARS-CoV-2 Neutralizing Antibody Detection Kit (B.1.617.1 Variant, Kappa; AdipoGen Life Sciences, San Diego, CA, USA) for the Kappa variant, and SARS-CoV-2 Neutralizing Antibody Detection Kit (B.1.617.2 Variant, Delta; AdipoGen Life Sciences) for the Delta variant. The absorbance of each well was measured using a Multiskan Fc with an incubator (Thermo Fisher Scientific, Waltham, MA, USA). All serum samples were frozen at −80 °C once after collection and were analyzed in duplicate.

### 2.4. Cytometric Bead Array

The level of serum cytokines was determined using a BD^TM^ Cytometric Bead Array Flex Kit (BD Biosciences, Franklin Lakes, NJ, USA) according to manufacturer instructions. All serum samples were diluted 1:6 and analyzed in duplicate. All data were analyzed on a BD FACS Verse Flow Cytometer (BD Biosciences) with FCAP Array Software v3.0 (BD Biosciences) according to manufacturer instructions.

### 2.5. Isolation of PBMCs

PBMCs were isolated using 8 mL of whole blood collected from COVID-19 patients after recovery and healthy volunteers using BD Vacutainer CPT tubes (BD Biosciences) according to the manufacturer’s instructions.

### 2.6. Single-Cell Sorting by Flow Cytometry

Isolated PBMCs were incubated with His-tagged SARS-CoV-2 S-RBD recombinant protein (Cell Signaling Technology, Danvers, MA, USA) for 30 min on ice. Cells were washed using phosphate-buffered saline (Nacalai Tesque, Kyoto, Japan) with 4% fetal bovine serum (Thermo Fisher Scientific) before staining with a fluorescence-conjugated antibody cocktail for 30 min on ice. The following antibodies were used; CD3 (SK7), CD4 (SK3), CD8 (RPA-T8), CD56 (B159), CD11b (D12), CD19 (HIB19), CD27 (O323), CD38 (HIT2), and human IgG (G18-145), all from BD Biosciences; FITC-conjugated His tag antibody from Abcam (Cambridge, UK); and anti-CD27 from BioLegend (San Diego, CA, USA). S-RBD-specific Bmems were defined as CD3^-^CD4^-^CD8^-^CD56^-^CD11b^-^CD19^+^CD27^+^IgG^+^His^+^ live cells. Single cells were sorted using a FACS AriaII instrument (BD Biosciences) into 96-well plates containing 4 μL/well of 8U RNAsin (Promega, Madison, WI, USA), 4U of recombinant RNase inhibitor (Takara Bio, Shiga, Japan), 10 mM DTT (Promega), and 10× PBS. All data were analyzed using the FlowJo software (TreeStar, San Carlos, CA).

### 2.7. Single-Cell RT-PCR and Immunoglobulin Gene Sequencing

IGHV- and IGLV-related genes were amplified using RNA ligase-mediated and oligo-capping rapid amplification of cDNA ends with the GeneRacer^TM^ Kit (Thermo Fisher Scientific) following manufacturer instructions, and the PCR product was obtained as described previously with slight modifications [34]. Briefly, the first step of PCR was performed according to the KOD FX Neo DNA polymerase guidelines (TOYOBO, Osaka, Japan). Nested PCR was performed in the same manner to improve specificity to the variable region, and after PCR purification using a QIAquick PCR Purification Kit (Qiagen, Hilden, Germany), the PCR product was sequenced using the reverse primer used for nested PCR using Sanger sequencing (Azenta Life Science, Tokyo, Japan). All primers were used as described in the Appendix A.

### 2.8. Statistical Analyses

A minimum of three independent experiments were included in each statistical analysis. Statistical significance was determined using the tests indicated in the figure legend. Statistical significance was set at *p* < 0.05. A standard curve was generated for each set of samples using a four-parameter logistic curve fit. All analyses were performed using the statistical programming software R version 4.0.3 (10 October 2020).

## 3. Results

### 3.1. Characterization of Serum Antibodies in COVID-19 Patients

We initially set out to investigate serum antibody features from patients who either exhibited ongoing COVID-19 or had recovered following infection during the second and third waves of infection in Japan between August 2020 and March 2021, prior to the emergence of the Alpha variant. In July 2021, serum samples were collected from recovered patients infected with SARS-CoV-2 VOCs encoding N501Y and L452R mutations. Upon admission to the hospital, oxygen saturation (SpO2) levels of moderate II patients infected with VOCs did not fulfill the criteria for disease severity (SpO2 ≤ 93%; Appendix A). This difference in disease severity may be because the symptoms of VOC-infected patients who were classified as those with moderate I disease on admission subsequently worsened and were then classified as those with moderate II disease. Serum samples from WT patients were analyzed for antibody binding to the S-RBD using enzyme-linked immunosorbent assay (ELISA) and anti-IgG and anti-IgM antibodies. We found that patients with moderate I and II illnesses exhibited higher IgG and IgM levels at discharge than at admission (Figure 1A,B). Next, we evaluated the serum samples for in vitro neutralizing antibody activity in blocking the binding between S-RBD WT and human ACE2. Serum-neutralizing antibody titer was positively correlated with serum anti-S-RBD IgM and IgG levels in all patients (Spearman’s correlation coefficient: 0.66, and 0.76, respectively; Figure 1C). The neutralizing antibody titer tended to be positively correlated with age in all the samples of the population (coefficient: 0.15) and appeared to be sex-independent (Figure 1D,E). We then examined the serum-neutralizing antibody titer from patients infected with VOCs via binding assays specific for the RBD domains of the WT, Kappa (formerly one of the variants of interest), and Delta strains. Serum-neutralizing antibody titer at discharge in patients infected with the WT strain increased with worsening disease severity (Figure 1F). Patients who eventually died exhibited low neutralizing antibody titer against the WT strain; however, one patient exhibited high titer against the Kappa and Delta variants (67.9% and 51.1%, respectively; Figure 1G,H). These data suggest that the generation of neutralizing antibodies against the infecting strain was necessary for recovery. Notably, we found that the polyclonal neutralizing antibody response in infected patients extended to strains beyond those responsible for the primary infection. Moderate II patients infected with the WT strain, excluding deceased patients, developed a higher neutralizing potency against the WT and variant strains than that observed in patients infected with VOCs containing the L452R mutation (Figure 1G,H). The mean value of serum-neutralizing antibody titer against the WT strain was 47.1% in WT-infected patients and 25.8% in VOC-infected patients (*p* = 0.044). For WT- and VOC-infected individuals, the rate was 53.6% and 25.4% against Kappa (*p* = 0.045) and 40.1% and 18.6% against Delta, respectively (*p* = 0.05; Figure 1G,H). Moreover, we divided VOC-infected patients into two groups of those infected with Alpha and Delta strains and found that the mean value of serum neutralizing antibody titer against the WT strain was 20.1% and 42.8%, those against Kappa were 19.8% and 42.1%, and those against Delta were 12% and 36%, respectively.

### 3.2. Serum Cytokine Features of COVID-19 Patients Infected with WT and VOC Strains

To characterize the cytokine response induced by different SARS-CoV-2 strains, we determined and compared the protein expression of 29 cytokines and chemokines in the sera of patients infected with WT and VOC strains at admission and discharge using a flow cytometric bead array (Appendix A). Upon discharge, serum levels of CCL2, a biomarker known to correlate positively with mortality risk in COVID-19 patients [35], trended slightly lower in mild and moderate I patients than in moderate II patients (Figure 2). Although CCL5 and CCL11 levels in patients infected with the WT strain tended to be higher at discharge than at admission, the CCL11 levels at discharge in patients infected with VOCs were higher than those at admission (*p* = 0.057) and much higher than those in moderate II patients infected with WT at discharge (*p* = 0.03). By contrast, serum levels of IL-6, CXCL10, and CXCL11 tended to be higher at admission than at discharge for all WT and VOC disease severities. Serum Fas ligand levels were inversely correlated with severity (Figure 2). In the two patients from our cohort who died during the experimental period, serum IL-6, IL-8, and CCL2 levels were higher than those of survivors (Appendix A; Appendix A).

Next, we compared serum cytokine levels upon admission between the WT and VOC-infected patient groups (Appendix A). The time from onset of symptoms to blood collection was similar (WT 8.3 ± 4.1 d, VOCs 8.0 ± 2.8 d). We found that the serum levels of IFN-α, IL-13, IL-10, and CCL11 were higher in VOC-infected patients than in WT-infected individuals, whereas IL-5 and IL-4 were higher in WT-infected patients than in those with VOCs. These data suggest that different immune responses occurred during infection with mutant and WT strains.

### 3.3. Correlation Analysis of Serum Cytokines and Neutralizing Potency in COVID-19 Patients

Next, we investigated the correlation between serum-neutralizing antibody titer against WT and serum cytokine levels over two-time points, including admission and discharge. Spearman’s correlation coefficient analysis showed a negative correlation between the titer and serum levels of most cytokines in all patients and within each severity group. Specifically, CXCL10, CXCL11, IL-6, and IL-10 serum levels showed strong negative correlations with neutralizing antibody titer in patients with moderate I and II disease (Figure 3A,B). This result is consistent with the reduction in acute inflammation caused by infection. Serum levels of granulocyte colony-stimulating factor (G-CSF) and granulocyte-macrophage colony-stimulating factor (GM-CSF) were negatively correlated with neutralizing antibody titer only in moderate II patients infected with VOCs. We found that the serum levels of CCL11 and IL-5 were positively correlated with neutralizing antibody titer, regardless of disease severity. Serum concentrations of IL-4 were positively correlated with the titer in patients with mild disease (Figure 3A). These factors are T-helper 2 (Th2) cytokines that contribute to allergic responses and antibody production, suggesting that elevated Th2 cytokine levels increase serum-neutralizing potency and alleviate symptoms [36]. Next, we assessed the correlation between patient age, serum cytokine level, and neutralizing antibody titer (Appendix A). Patient age was positively and significantly correlated with serum levels of IL-6, IL-8, MIG, and G-CSF (coefficients: 0.46, 0.40, 0.43, and 0.33, respectively). Patient age was negatively and significantly correlated with serum levels of Fas ligand (coefficient: −0.45), as was serum neutralizing antibody titer against WT (coefficient: −0.36). Furthermore, a strong positive correlation was observed between the serum levels of CXCL11, CXCL10, MIG, IL-6, IL-8, and IL-10 (Appendix A).

### 3.4. Distinct Immune Response in Recovered Patients and Naïve Individuals after mRNA Vaccination

To identify the changes in serum cytokine level associated with increased serum-neutralizing potency after mRNA-based COVID-19 vaccinations, we first investigated serum-neutralizing antibody titers in uninfected individuals and in those who recovered from WT infection. Sera were collected following their first and second vaccinations with mRNA-based vaccines BNT162b2 or mRNA-1273. WT-infected individuals had been infected 130 ± 5.4 d prior to vaccination. Sera were obtained from uninfected and WT-infected individuals at 14.0 and 17.3 ± 2.0 d after the first dose and at 24.5 ± 2.1 and 27.0 ± 6.7 d after the second dose, respectively. The serum-neutralizing antibody titer against the WT, Kappa, and Delta strains reached a peak after the first dose in the recovered patient group, except in one patient with mild disease (Figure 4A). Following the second dose, naïve and recovered patients exhibited significantly higher neutralizing antibody titers than those seen before vaccination, with a slightly stronger potency in recovered than in naïve patients (mean value: naïve 41.7%, recovered 57.8% against WT; naïve 61.1%, recovered 96.0% against Kappa; naïve 67.0%, recovered 95.7% against Delta; Figure 4A).

To investigate the role of cytokines in the mediation of immune responses following vaccinations, we performed a comprehensive analysis of 29 cytokine levels in the sera. The changes in cytokine levels after vaccination showed a specific trend, but significant differences were not observed because of the high interpersonal pre-vaccination variation in cytokine levels (Figure 4B). Specifically, serum IL-12p40 levels were elevated above pre-vaccine levels after the first dose and decreased after the second dose in both naïve and recovered patient groups. By contrast, the opposite was observed for serum CCL5 levels, which decreased after the first dose and increased after the second dose (Figure 4B). Serum levels of the Fas ligand, CXCL10, and CCL2 were slightly decreased after vaccination (Figure 4B). Interestingly, the mean serum levels of CXCL11 among naïve and recovered patient cohorts prior to vaccination were similar, but the recovered patients exhibited significantly higher CXCL11 serum levels post-vaccination.

We then examined whether correlations existed between serum neutralizing potency and cytokine level and among individual cytokines in naïve and recovered patients. We observed a slight positive correlation between IL-8 and IL-10 levels and neutralizing antibody titer against WT, Kappa, and Delta strains in recovered patients (coefficient: IL-8 vs. Kappa 0.55; IL-10 vs. WT 0.57; IL-10 vs. Kappa 0.55; IL-10 vs. Delta 0.57; Figure 4C). Although serum CXCL10 and CXCL11 levels were negatively correlated with the potency upon infection (Figure 3A and Appendix A), post-vaccination levels tended to be positively correlated in the recovered patient group (Figure 4C). Conversely, uninfected individuals showed a negative correlation trend similar to that of the recovered patient group upon infection. Across the entire group, serum levels of IL-8, IL-10, and CXCL11 were positively correlated with serum-neutralizing antibody titer (Appendix A). Post-vaccination IL-8 and Fas ligand levels were negatively correlated with each other in the naïve group and during initial infection of recovered patients but showed a positive correlation in recovered individuals (Figure 4C and Appendix A). Altogether, the difference in serum cytokine level changes between the naïve and recovered patient groups suggests that a unique immune response occurred after vaccination in recovered individuals.

### 3.5. Antigen-Specific Memory B-Cell Response in a COVID-19 Patient with High Neutralizing Potency after Vaccination

Long-term observation of a single group of patients for more than one year is needed to clarify the number of repeat COVID-19 vaccine doses required for recovered individuals. To determine the changes in serum-neutralizing potency in recovered patients following booster vaccination, we performed longitudinal analyses of serum-neutralizing potency and SARS-CoV-2-specific Bmem in peripheral blood following infection and vaccination. In our cohort, most patients showed an increase in serum-neutralizing potency at discharge, but one showed a drop. As all patients were initially infected and the time to admission from onset did not differ (data not shown), we excluded this patient from our cohort. To examine patients who have recovered with their neutralizing antibodies, we selected a representative healthcare worker infected with the WT strain who exhibited moderate II disease with a high neutralization rate upon infection (indicated by # in Figure 1F). This patient withdrew from treatment with an antiviral drug, favipiravir, because of its side effects and recovered only with nasal high-flow oxygen therapy. He was discharged on day 23 of hospitalization after a negative reverse transcription polymerase chain reaction (RT-PCR) test result. During the follow-up period, two vaccine doses were administered, one (BNT162b2) on day 273 and the second (BNT162b2) on day 297. The third booster dose (BNT162b2) was administered on day 495 after infection onset. We designated the time of medical examination as “before infection” and the time of appearance of symptoms upon infection as “day zero”. In this patient, serum IgM and IgG levels peaked at 10 and 20 d after symptom onset, respectively (Figure 5A). Serum-neutralizing antibody titer against the WT strain mirrored the IgG response, peaking at 20 d (inhibition rate: 55.4%) and decreasing to half its peak after 82 d (inhibition rate: 23.4%; Figure 5B). By day 185, the potency against the WT strain returned to the level observed before infection. Interestingly, serum-neutralizing antibody titer against the Delta variant gradually increased from day 10 post-infection and reached 44.7% on day 185 (Figure 5B). Fourteen days after the first vaccine dose, neutralizing antibody titer against the WT strain reached 55.8%, close to the peak at the time of infection, and reached a peak value of 7 d after the second dose of vaccination (Figure 5B). The potency gradually decreased to 19.0% after four months, and this level was maintained until the third vaccine dose (Figure 5B). By contrast, neutralizing potencies against Delta and Kappa variants peaked at 7 d after the first dose and remained high before and after the third vaccination (Figure 5B).

To investigate whether this fluctuation in serum neutralizing potency is associated with virus-specific Bmems, we performed fluorescence-activated cell sorting (FACS) analysis of peripheral blood mononuclear cells (PBMCs). The proportion of the IgG^+^CD19^+^CD27^+^ Bmem population increased after the second dose in the recovered individual. In particular, the CD19^+^CD27^+^ cell population showed a remarkable increase after the second and third vaccinations compared with that seen 272 d after onset (1.6- and 1.8-fold, respectively; Figure 5C,F). The percentage of Bmems binding to the recombinant S-RBD WT protein showed little change from onset to day 42 after the second vaccination but increased 2.86-fold after the third dose (Figure 5C). The absolute number of IgG^+^S-RBD^+^ Bmems and the percentage of lymphocytes were elevated after the first and third doses (Figure 5D,E). Notably, on day 272 from the onset, the proportion of IgG^+^S-RBD^+^ Bmems and CD38^high^ plasma cells was slightly higher than that on day 73, yet the serum neutralizing potency against the WT strain had almost disappeared (Figure 5B,C,F). These results suggest that serum-neutralizing potency did not coincide with the cellular dynamics of virus-specific Bmems.

### 3.6. Longitudinal Cytokine Profiles Reveal a Unique Immune Response after Repeated Vaccination

We found that intrapersonal neutralizing potency against the WT strain gradually decreased both after the initial infection and after vaccination. By contrast, serum-neutralizing potency against the Delta variant gradually increased after the initial infection and peaked after vaccination (Figure 5B). To gain insight into how immune responses mediate this phenomenon, we performed a longitudinal study on the effects of infection and mRNA-boosted vaccination on serum cytokine levels in the individual patient under long-term observation. Serum IL-6, IL-10, IL-12p40, G-CSF, CXCL10, and CXCL11 levels considerably increased starting on day 6 and diminished between days 10 and 20 (Figure 6A). CCL5 serum levels gradually increased and peaked by day 82, a pattern similar to that of serum-neutralizing antibody titer against the WT strain (Figure 5B and Figure 6A). On day 27, serum levels of IL-6 and IL-12p40 were transiently elevated. Intriguingly, serum levels of G-CSF remained high from day 27 to day 185. Notably, serum levels of IL-6, IL-8, IL-13, IL-12p40, CCL5, CXCL10, and CCL2 were markedly elevated for 3–7 d after each vaccination. We found that transient upregulation of CCL2 and CXCL10 expression was highest after the first dose, and repeated vaccination diminished this response (Figure 6A).

We then examined longitudinal serum cytokine levels from the patient and compared them against serum-neutralizing potency to identify potential correlations. We divided the time from the onset of symptoms to 185 days as an “infection” and from one day before the first dose to 36 days after the third dose as a “vaccination”. In the case of infection, serum neutralizing antibody titer against the WT strain exhibited a significant positive correlation with CCL5 (coefficient: 0.73, *p* = 0.02) and a negative correlation with G-CSF levels (coefficient: −0.69, *p* = 0.03; Figure 6B). In the case of vaccination, the potency against WT was significantly positively correlated with IL-10 levels (coefficient: 0.67, *p* = 0.009; Figure 6B). Correlations among individual cytokines differed between the infection and vaccination periods. Our findings that specific cytokines were released during vaccination and that transient elevations of CCL2 and CXCL10 levels were diminished by repeated vaccination suggest that the immune response may have been attenuated by repeat vaccination.

### 3.7. Affinity Maturation by the Third Booster Vaccination and the Natural Development That Occurred in Prolonged COVID-19 Patients Are Uniform

The relative percentage of Bmems binding to the recombinant S-RBD WT showed little change from onset to after the second vaccination, but the neutralizing potency against the WT strain showed transient upregulation (Figure 5B,C). To determine whether this change in serum neutralizing potency resulted from affinity maturation, IgG^+^S-RBD^+^ Bmems were single-cell sorted, and their antibody genes were sequenced. The distributions of both heavy and light chain antibody genes were skewed toward specific families on day 272 from the onset, but several types of antibody genes were observed (Figure 7A). After the second vaccination, a completely different gene family, immunoglobulin heavy chain variable (IGHV) 4-61*02/IGHV4-61*09, was selectively observed over 272 days in the heavy chain analysis (Figure 7A). Light chain analysis revealed that the minor gene family, immunoglobulin light chain (IGLV) 1-44*01, present at disease onset became a prominent family following the second vaccination (Figure 7B). Remarkably, new families emerged in the heavy and light chains following the third booster vaccination. These data suggest that booster vaccination elicited affinity maturation of S-RBD WT-specific Bmem antibody genes, consistent with the long-term maintenance of serum-neutralizing potency against VOCs in the recovered patient.

Finally, we examined the effect of prolonged infection on affinity maturation in an immunocompromised patient who developed COVID-19 pneumonia during rituximab maintenance treatment for follicular lymphoma, with lung lesions persisting for one year. Notably, the patient failed to develop anti-SARS-CoV-2 antibodies throughout the disease course (Appendix A), and the products used for immunoglobulin replacement therapy contained no anti-SARS-CoV-2 antibodies, suggesting that recovery from COVID-19 pneumonia was not due to patient-derived humoral immunity or external antibody supplementation [22,37]. To gain deeper insight into the effect of immunocompromise on affinity maturation, we performed sequencing of the antibody genes expressed during recovery in this patient and compared them to those from a WT-infected patient observed over time. Antibody-producing cells were isolated from PBMCs collected one year after infection. Although we had previously examined the antibody genes of IgG^+^S-RBD^+^ Bmems, we evaluated the CD19^+^CD27^+^IgG^+^ cell antibody gene because the S-RBD^+^ cell population was absent in the immunocompromised patient with prolonged COVID-19 (Appendix A). Intriguingly, we found that a major proportion of the IGHV4-61*02/IGHV4-61*09 and IGLV1-44*01 antibody gene families that emerged after the second vaccination, as well as a minor population comprising IGHV4-61*09 and IGLV1-44*01/IGLV1-44*03 that arose after the third dose in the recovered individual, were also present in the patient with prolonged COVID-19 (Figure 7A,B). To further characterize these antibody genes, we evaluated SHMs and the length of complementary-determining region 3 (CDR3). In the heavy and light chains of the recovered individual, the number of SHMs was significantly lower after the second or third vaccination than at 272 days from onset (*p* < 0.005; Figure 7C). In the immunocompromised patient with prolonged COVID-19, the number of SHMs in the heavy chain at one year from the onset was significantly lower than that at 272 days from the onset in the recovered individual but did not differ from post-vaccination levels in the recovered individual. By contrast, the SHM abundance in the light chain of the patient with prolonged COVID-19 did not differ at 272 days from the onset in the recovered individual but was significantly higher than that after the second or third vaccination (*p* < 0.005 and *p* = 0.007, respectively; Figure 7C). Although the length of CDR3 on the heavy chain was skewed after vaccination in the recovered patient, variable lengths were observed in the immunocompromised patient with prolonged COVID-19 (Figure 7D). By contrast, the light chain lengths in the recovered individual after the onset and vaccination and in the immunocompromised patient with prolonged COVID-19 trended toward 33 nucleotides (Figure 7D).

To investigate the immune response during infection in immunocompromised patients, we examined serum cytokine levels in a patient with prolonged COVID-19 at 17 and 71 d from the onset of infection. We found that IL-17F, which is involved in chronic inflammation and autoimmune diseases [38], was present in the serum, although it was not observed in other COVID-19 patients (Appendix A). These results suggest that this immunocompromised patient exhibited a distinct immune response upon infection compared with WT- or VOC-infected patients.

## 4. Discussion

The first highlight of this study is that IL-10 is a serum cytokine that correlates with post-vaccination neutralization potency in individuals who recovered from COVID-19. Moreover, we demonstrated that serum levels of CCL2, CXCL10, and IL-12p40 transiently increased immediately after vaccination. Notably, the increase in CCL2 and CXCL10 levels was attenuated by repeated vaccinations. Second, the distribution of the BCR repertoire after COVID-19 vaccination in recovered patients is similar to that of antibody gene families observed in a patient with long-term infection owing to immunosuppressive therapy with rituximab. The Bmem responses in this patient were similar to those seen after three-dose vaccination in healthy recovered participants. Although our findings supported the notion that immunocompromised patients do not produce anti-SARS-CoV-2 antibodies in their blood, they revealed evidence of affinity maturation in CD19^+^CD27^+^ B-cells with low IgG expression.

VOC-infected patients showed higher neutralizing potency against the WT strain than against VOCs, suggesting that the targets of induced neutralizing antibodies do not necessarily coincide with those of the infectious strain (Figure 1F–H). The Delta strain is more infectious and pathogenic to nasal epithelial cells than the ancestral strains, including the WT strain [8]. On the contrary, Miyauchi et al. found that the immune response to influenza viruses differs between vaccination and viral infection, and intranasal infection produces more broadly neutralizing antibodies than vaccination [39]. Taken together, VOCs, including the Delta strain, have more infectivity than the ancestral strains in the nasal epithelium, leading to the diverse neutralizing antibodies that recognize not only the S protein of Delta but also the common epitope of both WT and Delta strains in VOC-infected patients.

Robbiani D et al. showed that potent neutralizing antibodies exist even in patients with low or modest serum neutralizing activity [40]. We also demonstrated no correlation between serum-neutralizing potency and the percentage of memory B cells in the peripheral blood (Figure 5B–D). Thus, serum-neutralizing potency is independent of the quantity and quality of memory B cells. There are several reasons for this difference. First, high-quality antibodies with high neutralizing activity may be present in low amounts because they do not need to proliferate in the body. On the other hand, low-activity antibodies need to proliferate to enhance their neutralizing potency and occupy a major distribution in a single-cell analysis of approximately 100 cells. Second, monoclonal antibody cocktail therapy induces enhanced neutralizing activity with antibodies that recognize several epitopes [41]. The competition between antibodies of the same or similar epitopes may attenuate the effect of the original monoclonal antibody. Finally, infectivity-enhancing antibodies that bind to the N-terminal domain of the S protein are found in severe COVID-19 patients [42]. Thus, the neutralizing function does not necessarily coincide with binding to the target. We speculate that these discrepancies from serum neutralizing potency result from the induced antibodies that attenuate the effect of the potent neutralizing antibodies depending on the intrapersonal variation.

Immunocompromised hosts receiving convalescent plasma therapy, immunoglobulin replacement therapy, or antiviral drugs exhibit aberrant mutations in the SARS-CoV-2 genome that could trigger viral evolution [32]. As our immunocompromised patient underwent immunoglobulin replacement therapy for one year [22,37], suggesting that the viral genome may contain more mutations. The immunocompromised patient infected with the WT strain exhibited antibody gene sequences similar to those of post-vaccination antibodies in recovered patients with broadly neutralizing antibodies (Figure 7A,B). Furthermore, we identified antibody genes IGHV4-31*03 and IGHV4-59*01 in this patient and in moderate II patients infected with the Delta strain (data not shown), but not during infection or after vaccination in mild or moderate II patients infected with the WT strain. This is particularly interesting considering the value of therapeutic monoclonal antibodies because it suggests that immunocompromised patients generate broadly neutralizing antibodies against new variants created in the body. Collectively, our findings suggest that the determination of a few antibody gene sequences in recovered patients with immunodeficiencies is effective for producing clinical antibodies. The antibodies we discovered may exhibit broad neutralizing potency against unknown strains arising in the future.

To maximize the efficacy of therapeutics and vaccines, it is important to identify molecular biomarkers from the immune profile of each variant strain after infection or longitudinal dosing of the vaccine. We observed a markedly higher level of IFN-α upon admission in VOC-infected patients compared with that in WT-infected patients (Appendix A). IFN-α is a type I interferon, a type of cytokine involved in antiviral responses [43]. The variant, including the Delta strain, enhances pathogenicity more than the ancestral strain [8], probably resulting in elevated serum IFN-α levels in VOC-infected patients. We further found differences in serum levels of IL-4, IL-5, IL-10, and IL-13 between WT- and VOC-infected patients (Appendix A). These cytokines, mainly produced by eosinophils and basophils, contribute to Th2 responses involved in allergic reactions and antibody production [36,44], suggesting that the degree of Th2 response differs by the infectious strains. Pavel et al., demonstrated the interaction between IFN-γ-producing Th1 and Th2 in a high-risk group for severe COVID-19 disease, suggesting that Th1/Th2 cytokine imbalance is associated with higher mortality [45]. Several studies have shown that adjuvants, a key component of certain inactivated vaccines and antiviral drugs, induced robust and long-lasting specific immune responses, and those of inactivated coronavirus vaccines adjusting the Th1/Th2 balance are more effective without side effects [46]. Altogether, considering the attenuated efficacy of some antiviral drugs and therapeutic monoclonal antibodies against the Omicron strain [47,48], our findings suggest that the variants and unknown strains may be best treated using adjuvants that regulate the Th1/Th2 balance.

Takano et al., evaluated immune parameters, such as cytokine expression and cellular dynamics, that correlate with antibody response and systemic adverse events after two doses of an mRNA vaccine, revealing higher neutralizing potency and rates of systemic adverse events in a naïve patient group with high expression of IFN-γ, CCL2, CXCL9, and CXCL10 [49]. This finding is consistent with our results of a positive correlation between serum IL-10 level and neutralizing potency over 300 days, and with the observed transient upregulation of serum CCL2, CXCL10, and IL-12p40 levels from 3–7 d after each vaccination (Figure 6A,B). By contrast, we found distinct differences in serum cytokine levels associated with serum-neutralizing potency between naïve and recovered patients following vaccinations (Figure 4C). Given the asymptomatic infection that is one of the hallmarks of SARS-CoV-2, it is crucial to explore parameters in recovered patients, and our results support the possibility that IL-10 may be useful as a candidate biomarker.

This study had several limitations. First, our cohort comprised a small sample population. It is therefore impossible to determine whether the results observed by sequencing antibody genes in the immunocompromised patient was a unique characteristic of this patient or a phenomenon representative of immunocompromised patients in general. To this end, we must further investigate the antibody gene sequences of other immunocompromised patients who had COVID-19 and compare them with their post-vaccination sequences. Second, a cohort selection bias existed: inability to match age or time of hospitalization due to hospital conditions at the time of admission affected comparisons between groups. This resulted in differences in duration of hospitalization and age between WT- and VOC-infected patients with moderate II disease. It is important to use cohorts from multiple institutions matched for cofounders to precisely identify immune response differences by infectious strains. Third, we explored only the characteristic substitutions of each variant, without accounting for potential unexpected mutations. Our results showed distinct immune responses according to infectious strain, suggesting that the therapeutic regimen developed for the original SARS-CoV-2 strain may be less effective against variants. To maximize therapeutic efficacy, it is necessary to perform whole-genome sequencing and conduct detailed studies on infected strains and immune responses. Finally, we examined only the effects of mRNA vaccines. In a long-term analysis of inactivated vaccine recipients, the serum-neutralizing potency exhibits an increase after the second dose of the vaccine [50]. Specifically, the potency is higher in recovered patients than in naive individuals as well as those receiving the mRNA vaccine [51]. Hence, it is necessary to confirm whether the candidate biomarkers, such as IL-10, CCL2, CXCL10, and IL-12p40, described in this study also apply to inactivated vaccines. Collectively, our longitudinal, in-depth analyses of immune profiles in response to SARS-CoV-2 infection or vaccination, including characterization of the infected viral genome and sequencing of virus-specific antibody genes, will be useful to inform biomarker identification and novel therapeutic development to treat emerging SARS-CoV-2 variants. 

## 5. Conclusions

Our results showed that distinct immune responses occur depending on the viral strain and clinical history, suggesting that therapeutic options should be selected on a case-by-case basis. Furthermore, candidate biomarkers that correlate with repeated vaccination may support the efficiency and safety evaluation systems of mRNA vaccines and lead to the development of novel vaccine strategies.

## Figures and Tables

**Figure 1 vaccines-10-01815-f001:**
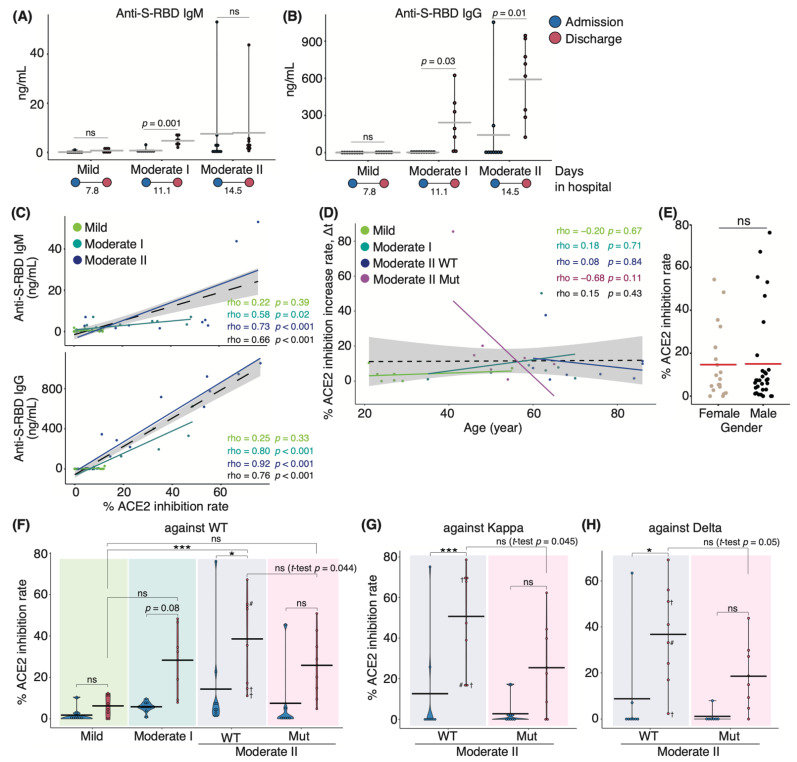
**Serum antibody features in COVID-19 patients during hospitalization:** Graphs show serum anti-S-RBD IgM (**A**) and IgG (**B**) levels for each disease severity (*n* = 9 patients/mild group; 10 patients/moderate I group; 9 patients/moderate II patients). Blue and red dots indicate admission and discharge, respectively; the duration of hospitalization is indicated at the bottom of each panel. Graphs show the correlation between anti-S-RBD IgM or IgG and neutralizing antibody titer (**C**) and age, as well as the change in the titer against the WT strain in patients infected with the WT and VOCs (**D**). Each dot denotes disease severity, and Spearman’s correlation coefficients are shown within the graph. Gray areas indicate a 95% confidence interval (CI) for the total. (**E**) Dot plot indicates the neutralizing potency against the WT strain by sex. Graphs show the serum-neutralizing antibody titer against WT (**F**), Kappa (**G**), and Delta (**H**) variants at each severity (moderate II is classified by the infected strains, *n* = 9 patients/mild group; 10 patients/moderate I group; 9 patients/WT-infected moderate II group; 8 patients/VOC-infected moderate II group). Statistical significances were determined using the Tukey–Kramer test and Welch’s *t*-test. All horizontal bars show mean values. Cross, deceased; #, long-term observed healthcare worker. * *p* < 0.05; *** *p* < 0.005; ns, not significant.

**Figure 2 vaccines-10-01815-f002:**
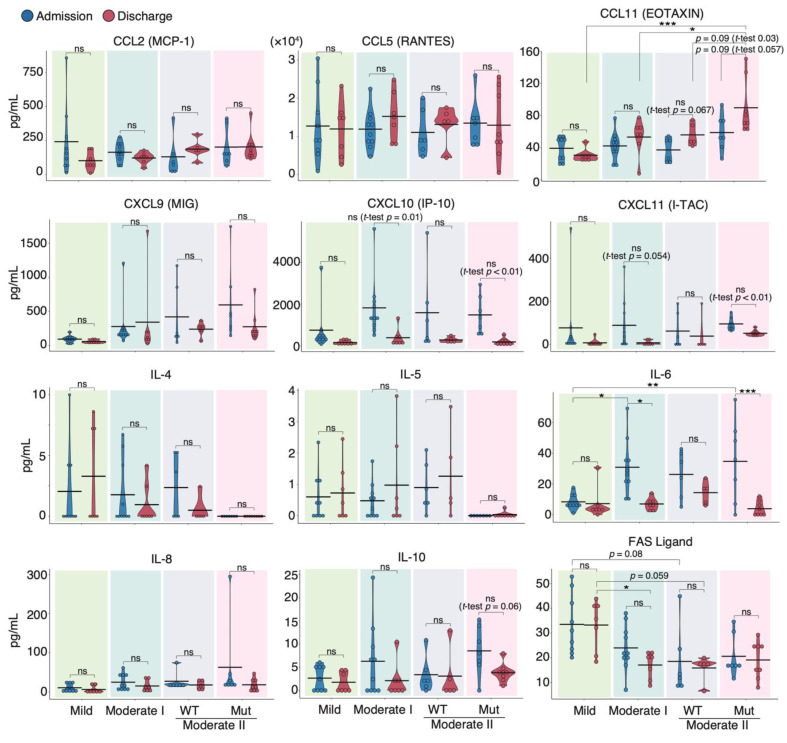
**Serum cytokine features of COVID-19 patients infected with the WT strain and VOCs.** Graphs show representative cytokine concentrations for each severity (*n* = 9 patients/mild group; 10 patients/moderate I group; 9 patients/WT-infected moderate II group; 8 patients/VOC-infected moderate II group). Blue and red dots indicate admission and discharge, respectively. Statistical significances were determined using the Tukey–Kramer test and Welch’s *t*-test. * *p* < 0.05; ** *p* < 0.01; *** *p* < 0.005; ns, not significant.

**Figure 3 vaccines-10-01815-f003:**
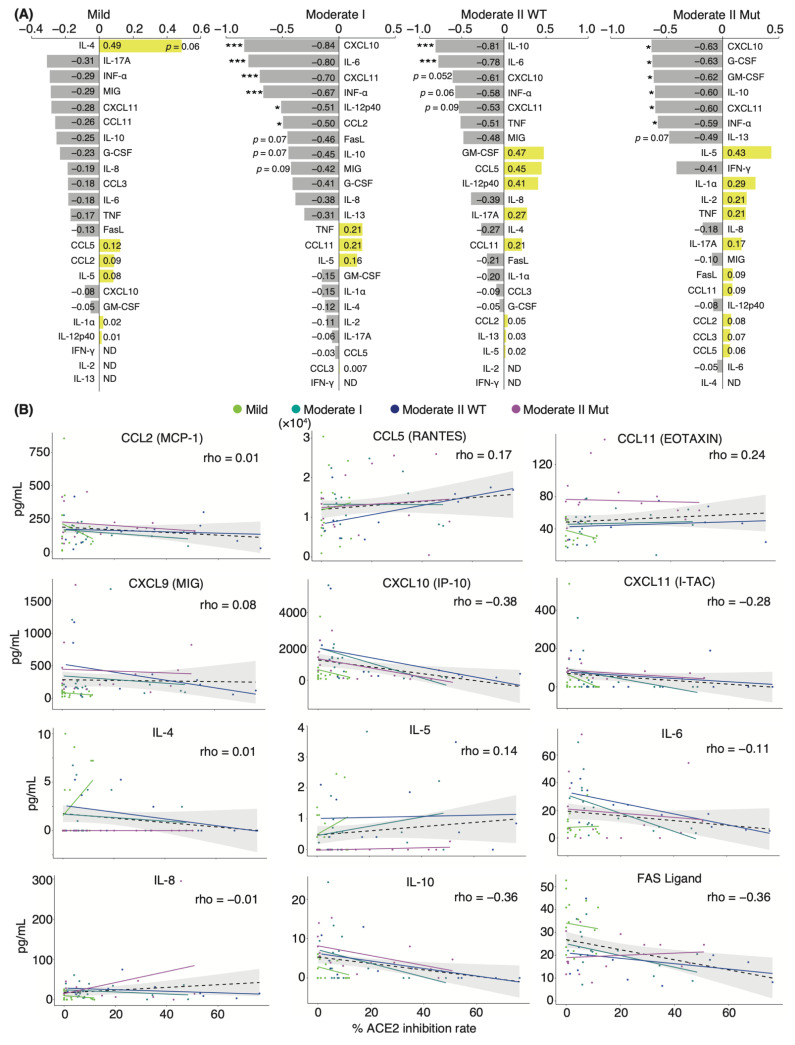
**Correlation analyses of serum cytokine levels and neutralizing antibody titer against WT in COVID-19 patients:** (**A**) Graphs show Spearman’s correlation coefficients of each severity between serum neutralizing antibody titer and individual cytokine levels (*n* = 9 patients/mild group; 10 patients/moderate I group; 9 patients/WT-infected moderate II group; 8 patients/VOC-infected moderate II group). (**B**) Graphs show representative correlation plots. Each dot denotes disease severity. Gray areas indicate 95% CI for the total. Statistical significances were determined using Tukey–Kramer test. * *p* < 0.05; *** *p* < 0.005; ND, not detected.

**Figure 4 vaccines-10-01815-f004:**
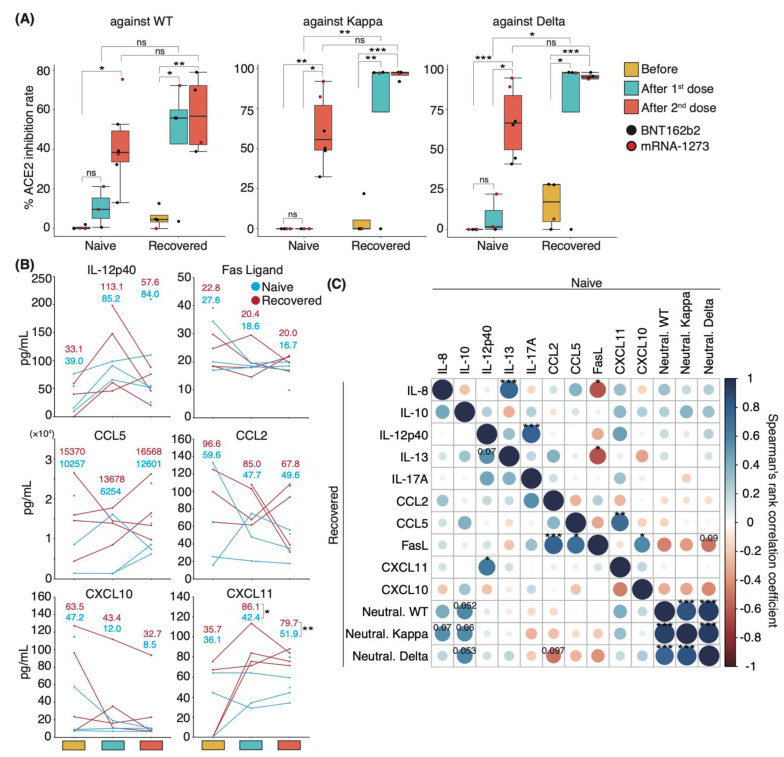
**Serum neutralizing potencies and cytokine levels in naïve and recovered individuals after mRNA vaccination:** (**A**) Box plots show neutralizing antibody titers against WT, Kappa, and Delta strains in naïve and recovered individuals after vaccination. Statistical significances were determined using Tukey–Kramer test. (**B**) Graphs show representative changes in serum cytokine levels during the vaccination period. Light blue indicates naïve, and red indicates recovered patients. Each mean is shown in the graph. (**C**) Spearman’s rank coefficients for naïve and recovered individuals are shown in an asymmetric correlation plot (*n* = 6 participants/naïve group; 4 participants/recovered group). The matrices across three time points for cytokine levels and neutralizing antibody titer, comparing naïve and recovered individuals. Neutral. WT, neutralizing potency for WT strain; Neutral. Kappa, neutralizing potency for Kappa strain; Neutral. Delta, neutralizing potency for Delta strain; * *p* < 0.05; ** *p* < 0.01; *** *p* < 0.005; ns, not significant.

**Figure 5 vaccines-10-01815-f005:**
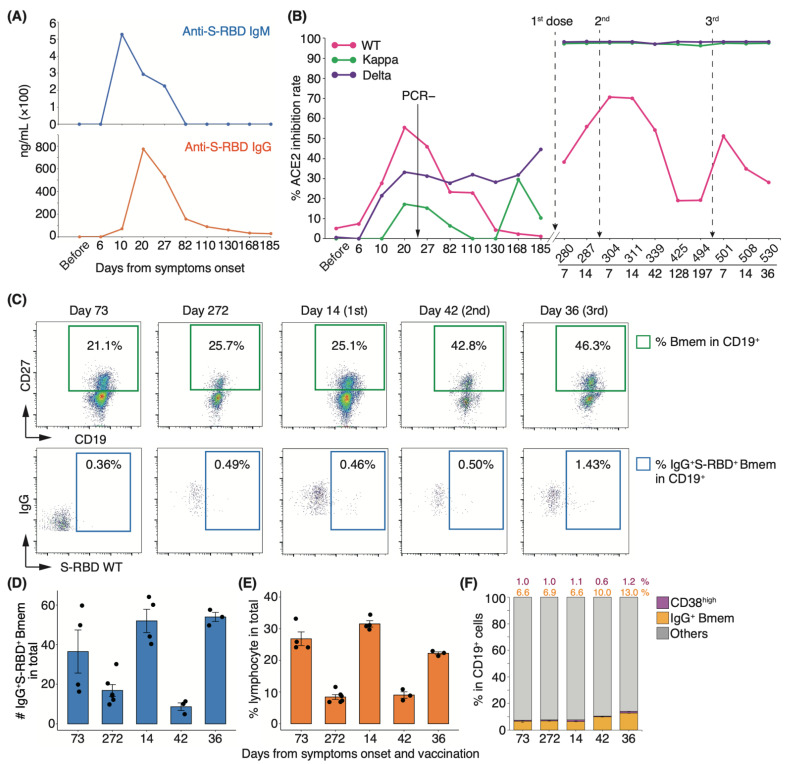
Longitudinal analyses of serum neutralizing antibody activities and PBMCs in a patient with high neutralizing potency: (**A**) Concentration of anti-S-RBD IgM and IgG in the serum of a healthcare worker (indicated by # in Figure 1) upon infection. (**B**) Fluctuation of each neutralizing antibody titer from onset to post-vaccination (*n* = 1). (**C**) Representative FACS plots of cell population are shown in panels. The numbers indicate the positive rate of each cell population. Graphs show the absolute number of IgG^+^S-RBD^+^ Bmems (**D**) and the frequency of lymphocyte (**E**), CD19^+^CD27^+^CD38^high^, and IgG^+^ Bmems ((**F**); *n* = 1, 3–5 technical replicates). Black dots indicate technical replicates. Each number indicates the mean.

**Figure 6 vaccines-10-01815-f006:**
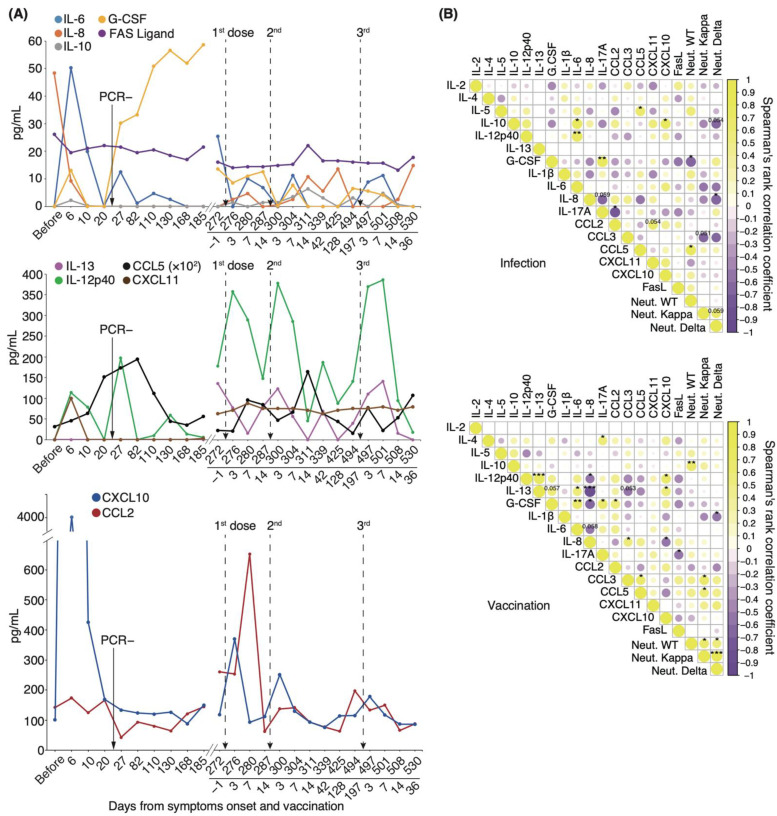
**Longitudinal analysis of serum cytokine levels associated with neutralizing potency:** (**A**) Representative cytokine, chemokine, and interleukin level fluctuation from onset to post-vaccination. (**B**) Correlation matrix on the left shows the time of infection from symptom onset to 185 days, and that on the right shows the vaccination period from 1 day before the first vaccination to 36 days after the third vaccination (*n* = 1). Neutral. WT, neutralizing potency for WT strain; Neutral. Kappa, neutralizing potency for Kappa strain; Neutral. Delta, neutralizing potency for Delta strain; * *p* < 0.05; ** *p* < 0.01; *** *p* < 0.005.

**Figure 7 vaccines-10-01815-f007:**
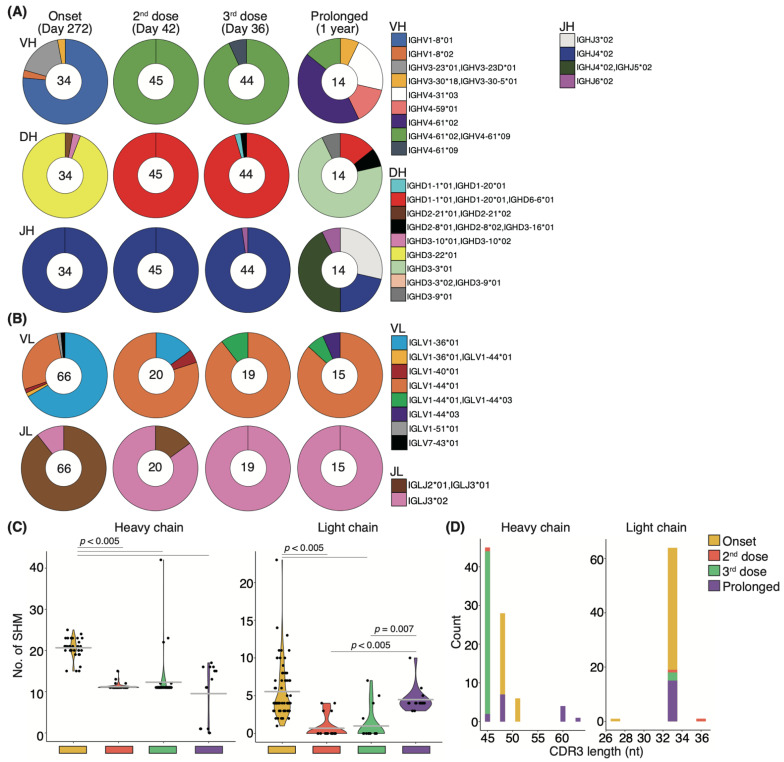
**Affinity maturation of S-RBD-specific memory B-cells after vaccination:** Pie charts show the distribution of heavy chain genes (**A**) and light chain genes (**B**) from onset to post-vaccination, comparing patients with the recovered individual (*n* = 1) and a prolonged COVID-19 patient (*n* = 1). The number in the inner circle indicates the number of sequenced clones. (**C**) Graphs show the number of somatic hypermutations in each antibody gene. Statistical significances were determined using Tukey–Kramer test. (**D**) Graphs show the distribution of CDR3 length in each antibody gene (heavy chains; *n* = 34–45 clones/recovered individual, 14 clones/prolonged COVID-19 patient; light chains; 18–66 clones/recovered individual, 15 clones/prolonged COVID-19 patient).

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
