# Peer review of "Longitudinal Analyses after COVID-19 Recovery or Prolonged Infection Reveal Unique Immunological Signatures after Repeated Vaccinations"

_vaccines, 2022, doi:10.3390/vaccines10111815_

Round 1

Reviewer 1 Report

Peer-review on manuscript:  Longitudinal analyses after COVID-19 recovery or prolonged infection reveal unique immunological signatures after repeated vaccinations” of Hisamatsu and co-authors.

The article under review provides molecular and cellular investigation of immune responses in infected, recovered, and vaccinated individuals. Neutralizing antibody activity against origin variant (WT) of  SARS-CoV-2 as well as against Kappa and Delta strains were evaluated in naive and recovered individuals after vaccination with 3 dose of BNT162b2 vaccine. Analysis of 29 cytokine levels and memory B-cell response in the sera was performed and their correlation with neutralizing antibody activity was assessed. The manuscript contains a huge amount of data valuable for improving understanding of the mechanisms of the immune response to SARS-CoV-2, as well as for therapeutic purposes.

Unfortunately, sometimes the material is presented not friendly towards the reader.

1) Figures in Supplementary Materials do not contain captions.

2) In the graphs of Fig. 5B, Fig. 6A, the time axis is broken, however, the graph is drawn continuously, which creates the illusion of an increase in the level of antibodies before vaccination

3) The arrangement of interleukins differs in numerous figures, which makes it very difficult for the reader to compare different results

4) The numbers of patients (which varies from one to several) are not indicated.

Author Response

Thank you for reviewing our manuscript entitled “Longitudinal analyses after COVID-19 recovery or prolonged infection reveal unique immunological signatures after repeated vaccinations (vaccines-1956122)”.

We are delighted to learn that our manuscript was met with much enthusiasm by the reviewers. We are now resubmitting our revised manuscript. We thank you and the reviewers for your thoughtful suggestions and insights, which have enriched the manuscript and produced a better and more balanced account of the research. We have addressed all the comments made by the reviewers, as indicated in the attached pages.

We hope that the revised manuscript is now suitable for publication in Vaccines. Thank you again for your consideration.

Sincerely,

Daisuke

Reviewer 2 Report

The study is both novel and rigorous. This is an meaningful study on cytokines and antibody levels memory B cell response in patients infected with the original SARS-CoV-2 strain or other variants of concern, and in vaccinated individuals, which complements data on long-term changes in antibody levels after vaccination.

Specific comments:

result

Figure 7, Davide F. Robbiani et al. In Convergent antibody responses to SARS-CoV-2 in convalescent individuals showed That “most convalescent plasma samples obtained from individuals who recover from COVID-19 do not contain high levels of neutralizing activity. Nevertheless, rare but recurring RBD-specifc antibodies with potent antiviral activity were found in all individuals tested”, the result of this article what's the difference?  

Discussion

Comparisons of dynamic changes in antibody levels after vaccination with other vaccines and inactivated vaccines could be added to the discussion.

Author Response

(The authors gave the same response as above.)

Reviewer 3 Report

In this article, Hisamatsu and colleagues evaluated whether serum cytokines correlated with neutralizing antibody activity in patients infected with various strains of SARS-CoV-2 and vaccinated individuals.

Major comments:

The study is largely descriptive, which is not necessarily a problem. However, there is a lack of solid conclusions that are made from the study. Furthermore, the article lacks clarity in both study design and aim throughout. The authors should consider more specifically stating which cytokines correlate with neutralizing antibodies in the abstract, and then state the meaning of this finding. Because the underlying data and methods are solid and describe important immune response signatures to SARS-CoV-2 infection and/or vaccination, it is my recommendation to re-consider the article following revision of the text that clarifies and specifies the primary findings from their studies.

Minor comments:

Pg. 2 line 55. “These variants exhibit increased infectivity and pathogenicity along with reduced sensitivity to neutralizing antibodies in comparison with the wild-type (WT) SARS-CoV-2 strain (3-8)”. The authors should define the strain of the virus that is wild-type here.

Author Response

We are grateful to Reviewer #3 for his/her critical comments and useful suggestions that have
helped lead us to important insights and considerably improve our paper.
Major comments:
The study is largely descriptive, which is not necessarily a problem. However, there is a
lack of solid conclusions that are made from the study. Furthermore, the article lacks
clarity in both study design and aim throughout. The authors should consider more
specifically stating which cytokines correlate with neutralizing antibodies in the abstract,
and then state the meaning of this finding. Because the underlying data and methods are
solid and describe important immune response signatures to SARS-CoV-2 infection
and/or vaccination, it is my recommendation to re-consider the article following revision
of the text that clarifies and specifies the primary findings from their studies.
We agree with the reviewer’s suggestion. We modified the abstract to state as follows.
Pg.1, line 28. “Longitudinal correlation analyses revealed that post-vaccination neutralizing
potential was more strongly associated with various serum cytokine levels in recovered patients
than in naïve individuals. We found that IL-10, CCL2, CXCL10, and IL-12p40 are candidate
biomarkers of serum neutralizing antibody titer after the vaccination of recovered individuals.”
Pg.1, line 35. “Candidate biomarkers that correlate with repeated vaccination may support the
efficacy and safety evaluation systems of mRNA vaccines and lead to the development of novel
vaccine strategies.”
Minor comments:
Pg. 2 line 55. “These variants exhibit increased infectivity and pathogenicity along with
reduced sensitivity to neutralizing antibodies in comparison with the wild-type (WT)
SARS-CoV-2 strain (3-8)”. The authors should define the strain of the virus that is wildtype
here.
We have described information regarding the wild-type (WT) mentioned within the references
or in this study as follows.

Pg.1, line 45. “However, these strategies have not changed since their development in 2019,
when the original SARS-CoV-2 strain (defined here as wild-type) emerged, despite the fact that multiple mutations have since been recorded”.

Thank you for reviewing our manuscript entitled “Longitudinal analyses after COVID-19 recovery or prolonged infection reveal unique immunological signatures after repeated vaccinations (vaccines-1956122)”.

We are delighted to learn that our manuscript was met with much enthusiasm by the reviewers. We are now resubmitting our revised manuscript. We thank you and the reviewers for your thoughtful suggestions and insights, which have enriched the manuscript and produced a better and more balanced account of the research. We have addressed all the comments made by the reviewers, as indicated in the attached pages.

We hope that the revised manuscript is now suitable for publication in Vaccines. Thank you again for your consideration.

Sincerely,

Daisuke

Reviewer 4 Report

The manuscript “Longitudinal analyses after COVID-19 recovery or prolonged

infection reveal unique immunological signatures after repeated vaccinations” systemically studied the cytokine and antibody profiles of COVID-19 patients, which is interesting and inspiring. It is a complete research paper to demonstrate thoroughly cytokine and antibody profiles after repeat vaccination and recovery from infection. However, there may still be some concerns as follows.

1.    Line 134-135: the cause dead patients should be variants of SARS-CoV-2, why did they have higher neutralizing antibody against variants?

2.    Figure 1F-H: authors should explain why the Mut-infected patients showed higher neutralization to WT virus than to Mut?

3.    Line 202-203: The allergy-related cytokines are positively correlated with neutralizing activity, authors should explain more about the relation.

4.     Line 325: authors should explain why or how or what rationale did the memory B cells not coincide with neutralizing activity.

5.    Did authors try to confirm the biomarkers that they have found in this study?

Author Response

(The authors gave the same response as above.)

Round 2

Reviewer 3 Report

The authors adequately clarified text and added more specific describing their results. Therefore, I recommend publication.